# Comprehensive Analysis of Ubiquitome Changes in *Nicotiana benthamiana* after Rice Stripe Virus Infection

**DOI:** 10.3390/v14112349

**Published:** 2022-10-26

**Authors:** Yu Liu, Chenyang Li, Yaqin Wang, Yi Xu, Jianxiang Wu, Xueping Zhou

**Affiliations:** 1State Key Laboratory of Rice Biology, Institute of Biotechnology, Zhejiang University, Hangzhou 310058, China; 2State Key Laboratory for Biology of Plant Diseases and Insect Pests, Institute of Plant Protection, Chinese Academy of Agricultural Sciences, Beijing 100193, China; 3Department of Plant Pathology, Nanjing Agricultural University, Nanjing 210095, China

**Keywords:** rice stripe virus, ubiquitomics, proteomics, *N. benthamiana*

## Abstract

Rice stripe virus (RSV) is one of the most devastating viruses affecting rice production. During virus infection, ubiquitination plays an important role in the dynamic regulation of host defenses. We combined the ubiquitomics approach with the label-free quantitation proteomics approach to investigate potential ubiquitination status changes of *Nicotiana benthamiana* infected with RSV. Bioinformatics analyses were performed to elucidate potential associations between proteins with differentially ubiquitinated sites (DUSs) and various cellular components/pathways during virus infection. In total, 399 DUSs in 313 proteins were identified and quantified, among them 244 ubiquitinated lysine (Kub) sites in 186 proteins were up-regulated and 155 Kub sites in 127 proteins were down-regulated at 10 days after RSV infection. Gene Ontology and Kyoto Encyclopedia of Genes and Genomes pathway enrichment analyses indicated that proteins with up-regulated Kub sites were significantly enriched in the ribosome. Silencing of 3-isopropylmalate dehydratase large subunit through virus-induced gene silencing delayed RSV infection, while silencing of mRNA-decapping enzyme-like protein promoted RSV symptom in the late stage of infection. Moreover, ubiquitination was observed in all seven RSV-encoded proteins. Our study supplied the comprehensive analysis of the ubiquitination changes in *N. benthamiana* after RSV infection, which is helpful for understanding RSV pathogenesis and RSV-host interactions.

## 1. Introduction

Rice stripe disease caused by rice stripe virus (RSV) is one of the most destructive virus diseases affecting rice production in East Asia [1]. RSV is a negative-sense and ambisense single-stranded RNA virus in the genus *Tenuivirus*. In the field, RSV is transmitted by the small brown planthopper (*Laodelphax striatellus* Fallén) and can infect a variety of monocotyledons, including rice, wheat and barley [2]. RSV can also be mechanically transmitted to *Nicotiana benthamiana* under laboratory conditions, which makes our research on RSV more convenient [3].

Ubiquitin is a polypeptide of 76-amino acid which is highly conserved in eukaryotic organisms. Ubiquitination is an essential post-translational modification (PTM) that covalently conjugates single (monoubiquitin) or multiple (polyubiquitin chain) ubiquitins to one or more lysine residues of a target protein [4]. This process requires the sequential action of three enzymes: ubiquitin-activating enzyme (E1), ubiquitin-conjugating enzyme (E2), and ubiquitin ligase (E3) [5]. In most cases, ubiquitin acts as a molecular label within the ubiquitin-proteasome system (UPS), which mediates the degradation of cellular proteins [5]. However, ubiquitination also involves in a variety of cellular processes, including DNA damage repair, DNA replication, receptor-mediated endocytosis and innate immune signals [6,7].

Ubiquitination plays an important role in the dynamic regulation of host defense in the process of virus infection. Hosts can degrade virus-encoded proteins through ubiquitination to resist virus infection. For example, tobacco RING-finger protein (NtRFP) interacts with a geminivirus βC1 to mediate its ubiquitination and degradation via the UPS [8]. Viruses, in turn, can disrupt the ubiquitination pathway by interacting and interfering with key elements in the ubiquitination pathway, and even utilize this pathway to degrade antiviral proteins of hosts via interacting with E3 ligases, thus promoting their own infection. For example, two viral suppressors of RNA silencing P0 and P25 encoded by a polerovirus, and potato X virus (PVX) can interact with and degrade Argonaute1 (AGO1) through the formation of SKP1, CUL1, F-box protein (SCF) complex with host factors [9,10]. However, P0 leads AGO1 into the autophagic lysosomal pathway [9], while P25 guides AGO1 into the 26S proteasome pathway for degradation [10].

A novel mass spectrometry (MS)-based diglycine (di-Gly) proteomics approach has been developed to identify lysine-ubiquitinated proteins with the development of an antibody that could specifically recognize the di-Gly-Lys remnant and enrich ubiquitinated material following trypsin digestion [11,12,13,14]. With the development of mass spectrometry, proteomics have emerged as a core component of fundamental discoveries in virology in recent years [15,16]. However, there have been little reports about the ubiquitome of plants infected with viruses. In this study, we conducted ubiquitomic analyses by using a label-free mass spectrometry-based analysis of protein ubiquitination with the di-Gly-Lys–remnant antibody enrichment approach to investigate the global changes of potential ubiquitination status of *N. benthamiana* infected with RSV. 399 differentially ubiquitinated sites (DUSs) in 313 proteins were identified and quantified, among which 244 Kub sites in 186 proteins were up-regulated and 155 Kub sites in 127 proteins were down-regulated at 10 days after RSV infection. Bioinformatics analyses were performed to elucidate potential associations between differentially ubiquitinated proteins and various cellular components/pathways during RSV infection. Our study provides new information for understanding RSV pathogenesis and RSV-host interactions.

## 2. Materials and Methods

### 2.1. Plant Materials, Agroinfiltration, and Virus Inoculation

Five-leaf-old *N. benthamiana* plants were used for agroinfiltration as described previously [17]. For protein transient expression in *N. benthamiana* leaves, the concentration of *Agrobacterium tumefaciens* was adjusted to OD_600_ = 0.6. For virus-induced gene silencing (VIGS), the OD_600_ of pTRV1 and pTRV2 was 0.1. RSV mechanical inoculation on *N. benthamiana* by using RSV infected rice leaves was performed as described previously [18].

For omics analyses, sixty *N. benthamiana* plants were divided into 6 groups, each group contained 10 plants. Three groups were inoculated with RSV, while the other 3 groups were mock-inoculated. The upper 3 leaves were harvested and stored in −80 °C freezer at 10 days post inoculation (dpi).

### 2.2. Plasmid Construction

The RNA of healthy or RSV-infected *N. benthamiana* was isolated by using RNAprep Pure Plant Plus Kit (cat: DP441, TIANGEN, Beijing, China) and then reverse transcripted to cDNA by using ReverTra Ace^®^qPCR RT Master Mix with gDNA Remover (cat: FSQ-301, TOYOBO, Osaka, Japan) according to the manufacturer’s instructions. DNA fragments of coding sequences of eGFP, RSV non-structure protein 2 (NS2), RSV non-structure protein 3 (NS3) and RSV coat protein (CP) were cloned by PCR amplification with respective primers (Appendix A) and then iused into the ClonExpress II One Step Cloning vector (Vazyme, Nanjing, Jiangsu, China) according to the manufacturer’s instructions. The *in planta* expression vector pGD-2×35S-MCS-3×Flag was digested by *Xho*I and *Sac*I, coding sequences of eGFP, RSV NS2, RSV NS3 and RSV CP were then fused into the vector. For tobacco rattle virus (TRV) based VIGS, 300 bp fragments of 3-isopropylmalate dehydratase large subunit (NbLeuC), epoxide hydrolase 3 (NbEPHX3) and mRNA-decapping enzyme-like protein (NbDCP1) were cloned by PCR amplification with respective primers (Appendix A) and then fused into the pTRV2 vector that digested by *Xba*I and *Xho*I, respectively.

### 2.3. Western Blotting and Antibodies

The extraction of total protein of *N. benthamiana* and western blotting were performed as described previously [19]. Specific primary antibodies against Ubiquitin (cat: PTM-1106, PTM Bio, Hangzhou, Zhejiang, China), Flag tag (cat: F1804, Sigma-Aldrich, St. Louis, MO, USA), ribulose bisphosphate carboxylase large chain (RbcL) (cat: ER1919-16, HUABIO, Hangzhou, Zhejiang, China) and Actin (cat: AC009, ABclonal, Wuhan, Hubei, China) were used in western blotting. Primary antibody that against RSV CP were generated in our lab as described previously [20].

### 2.4. Protein Extraction and Trypsin Digestion

The protein extraction and subsequent trypsin digestion for omics analyses were performed as described previously [21]. In brief, the *N. benthamiana* leaves were grinded by liquid nitrogen into powder and then dissolved in lysis buffer (8 M urea, 1% Triton-100, 10 mM dithiothreitol, and 1% protease inhibitor cocktail). After centrifugation at 20,000× *g* at 4 °C for 10 min, the leaf debris was removed and the proteins in the supernatant were precipitated with cold 20% TCA for 2 h at −20 °C. The precipitate was collected by centrifugation at 12,000× *g*, 4 °C for 10 min. After being washed with cold acetone, the precipitate was then redissolved in 8 M urea.

For trypsin digestion, 5 mM dithiothreitol was used to reduce the protein solution and then the protein solution was alkylated with 11 mM iodoacetamide. The protein sample was diluted with 100 mM NH_4_HCO_3_ till the urea concentration less than 2 M. Finally, the protein sample was trypsin digested at 1:50 trypsin-to-protein mass ratio overnight and 1:100 trypsin-to-protein mass ratio for 4 h.

### 2.5. Affinity Enrichment of Ubiquitinated Peptides

Affinity enrichment of specific peptides was performed as described previously [22]. Anti-diglycine lysine antibody conjugated agarose beads (cat: PTM-1104, PTM Bio, Hangzhou, Zhejiang, China) were used for affinity enrichment of peptides that contained glycine-glycine remnants of ubiquitinated lysine (K-ε-GG) after trypsin digestion.

### 2.6. Proteome and Ubiquitome Analysis by LC–MS/MS

Ultra-performance liquid chromatography (UPLC) was performed as described previously [21]. EASY-nLC system (Thermo Scientific, Waltham, MA, USA), 0.1% formic acid (solvent A) and 0.1% formic acid in 90% acetonitrile (solvent B) were used for UPLC. Briefly, for the proteome analysis, the liquid phase gradient was set as 0 min, 100% A; 0–160 min, 5–25% B; 160–210 min, 25–40% B; 210–225 min, 40–80% B; 225–240 min, 80% B. The flow rate was set to 300 nL/min. For the ubiquitome analysis, the liquid phase gradient was set as 0 min, 100% A; 0–60 min, 2–25% B; 60–78 min, 25–40% B; 78–84 min, 40–80% B; 84–90 min, 80% B. Flow rate was set to 350 nL/min.

The peptides were then subjected to a nanospray ion (NSI) source followed by tandem mass spectrometry (MS/MS) in Q ExactiveTM Plus (Thermo Scientific). The parameters of MS/MS were set as described previously [21].

### 2.7. Database Search

Maxquant search engine (v.1.5.2.8) was used for MS/MS data processing as described previously with a little change [23]. The *N. benthamiana* database V1.0.1 (https://solgenomics.net/ftp/genomes/Nicotiana_benthamiana/ (accessed on 13 July 2017)) and RSV encoded protein sequences (GenBank: ABC68333.1, ABC68334.1, ABC68335.1, ABC68336.1, ABC68337.1, ABC68338.1, ABC68339.1) concatenated with reverse decoy database were used for MS/MS data search. Besides, label-free quantification (LFQ) and unique peptides for quantification were selected. For the proteome data, up to 2 missing cleavages were allowed for Trypsin/P. As to the ubiquitome data, up to 4 missing cleavages were allowed and ubiquination on lys was specified as variable modification. Perseus (v1.5.3.2) was used to impute missing values by normal distribution (width = 0.3; shift = 1.8) assuming that these proteins were near to the detection limit. After missing value imputation, the proteome data were used to normalize the ubiquitome data.

### 2.8. Bioinformatics Analysis

The model of 21-mers sequences that modified by ubiquitin (10 amino acids up- and down-stream of the Kub site) was analyzed with Software Motif-x [24]. All the database protein sequences were used as background database parameter, other parameters with default. The Proteins with DUSs were annotated with Gene Ontology (GO), Kyoto Encyclopedia of Genes and Genomes (KEGG), protein domain and subcellular localization as described previously [23]. UniProt-GOA database (http://www.ebi.ac.uk/GOA/ (accessed on 31 August 2017)) and InterProScan (http://www.ebi.ac.uk/interpro/ (accessed on 31 August 2017)) were used for GO annotation. KAAS (http://www.genome.jp/kaas-bin/kaas_main (accessed on 31 August 2017)) and KEGG Mapper (http://www.kegg.jp/kegg/mapper.html (accessed on 31 August 2017)) were used for KEGG annotation. InterProScan was used for protein domain annotation. Wolfpsort (http://www.genscript.com/psort/wolf_psort.html (accessed on 31 August 2017)) and CELLO (http://cello.life.nctu.edu.tw/ (accessed on 31 August 2017)) were used for subcellular localization annotation. Perl module (https://metacpan.org/pod/Text::NSP::Measures::2D::Fisher (accessed on 31 August 2017)) was used for functional enrichment analysis. For each category, a two-tailed Fisher’s exact test was used to test the enrichment of the proteins with differentially ubiquitinated sites (DUSs) against all identified proteins. The significance threshold was set at *p* value < 0.05.

### 2.9. Statistical Analysis

Every result was presented as the mean ± SD. An unpaired *t*-test was used to determine the significance of the differences between RSV-infected groups and Mock groups. The significance threshold was set at *p* value < 0.05.

### 2.10. Real-Time Quantitative PCR

Real-time quantitative PCR (RT-qPCR) was performed as described previously [25]. Oligo 7 software (https://www.oligo.net/ (accessed on 21 September 2021)) was used to design primers for RT-qPCR. Actin was used as a reference gene for relative quantification. Sequences of primers for RT-qPCR were provided in Appendix A.

### 2.11. Immunoprecipitation Assay

4-week-old *N. benthamiana* leaves were used for proteins expression via agroinfiltration. IP buffer that contained 40 mM Tris-HCl (pH 7.5), 150 mM NaCl, 2 mM EDTA, 5 mM MgCl_2_, 5 mM DTT, 5% glycerol, 0.1% Triton X-100, and protease inhibitor cocktail (cat: HY-K0011, MedChemExpress, Monmouth Junction, NJ, USA) was used for protein extraction. The debris of *N. benthamiana* leaves was removed through centrifugation at 16,000× *g*, 4 °C for 15 min twice and the supernatant was incubated with Anti-Flag M2 magnetic beads (cat: M8823, Sigma-Aldrich, St. Louis, MO, USA) at 4 °C for 2 h. Then beads were washed with IP buffer till no green residue in IP buffer. Finally, proteins that immunoprecipitated with beads were denatured with loading buffer and follow by western blotting analysis.

## 3. Results

### 3.1. General Features of the Quantitative Proteome and Ubiquitome Datasets

To reveal the global changes of the host ubiquitination status due to RSV infection, a relative quantitative ubiquitomics approach was performed using leaves from RSV-inoculated and mock-inoculated *N. benthamiana* at 10 dpi (Figure 1A). Either RSV-inoculated or mock-inoculated *N. benthamiana* plants possess three biological replicates. Six groups of leaves were harvested and prepared for western blotting analysis to confirm an effective infection (Figure 1B). To obtain the ubiquitome data, we enriched peptides that contained glycine-glycine remnants of ubiquitinated lysine (K-ε-GG) after trypsin digestion by using of specific antibody and then identified and quantified them through LC−MS/MS combining with LFQ (Figure 1C). To eliminate the influence of protein expression changes, a matching proteome analysis was performed at the same time; all ubiquitome values were normalized with the corresponding protein abundance values (Figure 1C).

Analysis of the proteome data from RSV-infected and mock leaves identified 5519 proteins, of which 3549 of these proteins were quantifiable (Table 1). Based on the 1.5-fold change and the *t*-test *p* value less than 0.05 as the significance threshold, 323 proteins were up-regulated and 276 were down-regulated after RSV infection (Table 2). Analysis of the ubiquitome data identified 2530 Kub sites in 1469 proteins, among which 1345 Kub sites in 862 proteins were quantifiable (Table 1). The proteome data and ubiquitome data were subjected to missing-value imputation before the proteome data was used to normalize the ubiquitome data [26,27]. In total, we obtained 399 DUSs in 313 proteins, among which 244 Kub sites in 186 proteins were up-regulated and 155 Kub sites in 127 proteins were down-regulated after RSV infection (Table 2 and Appendix A).

### 3.2. Motif Analysis of Kub Sites

Ten amino acids up and downstream of Kub site were analyzed to distinguish the pattern of sequences around the Kub site. Of the ubiquitinated peptides identified, 11 conserved Kub motifs were found: KubG, AKKub, GKub, RKub, ExxxxxxKKub, KKub, SKub, KubxG, AKub, KubxA and ExxKub (x: a random amino acid residue; G: glycine; A: alanine; R: arginine; E: glutamic acid; S: serine) (Figure 2A). Detailed information was listed in Appendix A. A heat map was generated to show the enrichment or depletion of amino acids surrounding the Kub site (Figure 2B). G in the ±1 and +2 positions, K and R in the −1 position and A in the +1 position were significantly abundant, while K in the +1 to +4 positions, isoleucine (I), leucine (L) and proline (P) in the −1 position and R in the +1 position were depleted.

### 3.3. Functional Classification of Proteins with DUSs

GO analysis was performed to examine the potential biological functions of proteins with DUSs induced after RSV infection. Proteins with up- and down-regulated Kub sites were classified into three categories (“biological process”, “cellular component”, and “molecular function”) respectively (Figure 3A,B). Proteins with up- and down-regulated Kub sites were associated with similar terms (Figure 3A,B). In the biological process category, the most of proteins with DUSs were associated with metabolic process. In the cellular component category, proteins were mainly participated in cell. In the molecular function, the majority of proteins with DUSs were involved in binding.

Subcellular location classification revealed that proteins with DUSs were mainly localized in cytoplasm, chloroplast and nucleus (Figure 3C,D).

### 3.4. Functional Enrichment of Proteins with DUSs

For the GO enrichment analysis, in the “biological process” category, the most enriched terms for proteins with up- and down-regulated Kub sites were “peptide metabolic process” (Figure 4A) and “oxoacid metabolic process” (Figure 4B), respectively; while in the “cellular component” category, the term “ribosome” ranked first for proteins with up-regulated Kub sites (Figure 4A), and proteins with down-regulated Kub sites were enriched mostly in “anchored component of membrane” and “anchored component of plasma membrane” (Figure 4B). Within the “molecular function” category, proteins with up-regulated Kub sites were enriched mostly in “structural constituent of ribosome” and “structural molecule activity” (Figure 4A). Proteins with down-regulated DUSs were enriched in “magnesium ion binding” and “unfolded protein binding” (Figure 4B).

KEGG pathway enrichment analysis showed that proteins with up-regulated Kub sites were mainly enriched in the pathway “ribosome” (Figure 4C). Proteins with down-regulated Kub sites were mainly enriched in the pathway “glycolysis/gluconeogenesis” (Figure 4D).

Protein domain enrichment analysis of proteins with DUSs showed that the domain “heat shock protein Hsp90, N-terminal” ranked first for proteins with up- and down-regulated Kub sites (Figure 4E,F).

The ribosome related terms emerged multiple times in the functional enrichment analysis of proteins with up-regulated Kub sites. Direct analysis of ribosome pathway showed that 17 ribosome proteins possess up-regulated Kub sites (Appendix A).

### 3.5. Influence of Host Proteins with DUSs on RSV Infections

To investigate the potential functions of host proteins with DUSs during RSV infection, three proteins that possessed significantly regulated Kub sites were selected for VIGS based on TRV (Table 3). All the three chosen proteins possess only one identified Kub site. The ubiquitination change of Kub site could be simply used to represent the ubiquitination change of the corresponding protein. Among these three proteins, the ubiquitination levels of NbLeuC and NbEPHX3 were significantly down-regulated and the ubiquitination level of NbDCP1 was significantly up-regulated. Compared with the healthy *N. benthamiana*, plants infected with TRV-GFP, NbLeuC and NbEPHX3-silenced plants displayed normal growth, but the NbDCP1-silenced plants showed slightly delayed growth (Figure 5A). RT-qPCR showed that the TRV based VIGS system efficiently down-regulated the target gene expression at 10 dpi (Figure 5B). Then, NbLeuC-silenced, NbEPHX3-silenced, NbDCP1-silenced, and TRV-GFP control plants were inoculated with RSV. NbLeuC-silenced plants displayed milder RSV symptoms compared with TRV-GFP control plants at 15 dpi (Figure 5C). However, NbEPHX3-silenced plants and NbDCP1-silenced plants displayed symptoms similar to those of TRV-GFP control plants at 15 dpi (Figure 5C). Consistent with virus symptoms, western blotting analysis showed that the accumulation of the RSV CP in NbLeuC-silenced plants was significantly lower than that in TRV-GFP control plants (Appendix A). However, the accumulation of the RSV CP protein in NbEPHX3-silenced and NbDCP1-silenced plants were equal to that in TRV-GFP control plants (Appendix A). Interestingly, NbDCP1-silenced plants displayed severe symptoms compared with TRV-GFP control plants at 30 dpi (Figure 5D). The accumulation of the RSV CP in NbDCP1-silenced plants was equal to that in TRV-GFP control plants at 30 dpi (Appendix A). The severe chlorosis in NbDCP1-silenced plants meant that the chloroplasts might be seriously damaged. The accumulation of chloroplast protein RbcL was detected and showed a significant decrease compared with TRV-GFP control plants, which was consistent with the chlorosis symptom (Appendix A).

### 3.6. Ubiquitination of RSV Proteins

The ubiquitination of viral proteins has great impact on viral pathogenesis [8]. Here, besides the host protein, we also analyzed the ubiquitination of RSV proteins. All seven RSV proteins were identified in the proteome data. The sequence coverage was listed in Table 4. At least 1 Kub residue (RdRp) and up to 11 Kub sites (CP) were identified in RSV proteins (Table 4). To confirm the LC–MS/MS result, *A. tumefaciens*-delivered plasmids engineered to express the Flag epitope-tagged eGFP, NS2, NS3 and CP were separately infiltrated into *N. benthamiana* leaves and immunoprecipitation assays were performed. Western blotting showed that eGFP-Flag, NS2-Flag, NS3-Flag and CP-Flag were effectively enriched with the anti-Flag magnetic beads. Bands of higher molecular mass were only detected with the antibody against ubiquitin in the IP samples of NS2-Flag, NS3-Flag and CP-Flag (Figure 6), suggesting that these three viral proteins were ubiquitinated.

## 4. Discussion

Ubiquitination has been extensively shown to play important roles during RSV infection. Ubiquitin-conjugating enzyme E2 E inhibits the accumulation of RSV in *Laodelphax striatellus* [28]. RSV NS2 and NS3 could interfere with the function of 26S proteasome and protect ubiquitinated proteins from degradation in *L. striatellus* via direct interaction with RPN8 and RPN3, respectively [29,30]. The ovarian tumor (OTU) protease domain in RdRp of RSV possesses deubiquitinating enzyme (DUB) activity in vitro and in insect cells [31]. Ubiquitin like protein 5 (UBL5) mediates the degradation of RSV NS3 through the 26S proteasome via direct interaction but not the formation of covalent conjugate in *N. benthamiana* [32]. Microtubule-associated C4HC3-type E3 Ligase (MEL) responses to RSV infection and initiates a series of host immune signaling by mediating SHMT1 degradation through the 26S proteasome pathway in *N. benthamiana* and *Oryza sativa* [33]. The emergence and development of ubiquitomics provide effective technical support for us to explore the unprecedented details into biological processes controlled by ubiquitin and its interplay role during RSV infection [11,34].

An increasing number of studies about ubiquitome of various species have been reported since the emergence of the antibody that could recognise the di-Gly-Lys remnant [13]. Some of these studies have reported that the ubiquitome of organisms can respond to abiotic or biotic stresses, such as drought stress, ultraviolet-B irradiation and virus infection [23,24,35]. But most of them only quantify the ubiquitome changes simply with the K-ε-GG peptide abundance, take no account of the influence of protein expression changes. It should be noted that an increase in the MS quantitation value of a K-ε-GG peptide is more commonly explained by an increase in the abundance of its corresponding protein [36,37]. In our study, through LFQ technology and K-ε-GG based qualitative method, we simultaneously performed proteomic and ubiquitomic analysis to investigate the ubiquitination change after RSV infection of *N. benthamiana* plant. Combined with the TRV-mediated VIGS technique, our research may help us to illuminate the molecule mechanisms upon RSV infection.

In total, we identified 399 DUSs, among which 244 Kub sites were up-regulated and 155 Kub sites were down-regulated during RSV infection. GO, KEGG and Protein domain enrichment analyses were performed to investigate enrichment of proteins in functional categories. Comprehensive analyzation indicates that the ubiquitination of plentiful ribosome related proteins was significantly up-regulated (Appendix A). Many of them are accompanied with down-regulated protein expression which means that RSV infection might lead to the ubiquitination and degradation of ribosome related proteins and eventually interfere the normal protein biosynthesis.

Three proteins were selected to investigate the potential functions of proteins with DUSs during RSV infection. NbLeuC and NbEPHX3 displayed down-regulated ubiquitination and up-regulated protein expression, which might be more stable during RSV infection. LeuC catalyzes the isomerization between 2-isopropylmalate and 3-isopropylmalate, via the formation of 2-isopropylmaleate which is part of the pathway L-leucine biosynthesis in *Arabidopsis thaliana* [38]. EPHX3 catalyzes the hydrolysis of epoxide-containing fatty acids in human cells [39]. So far, no research has shown that LeuC is associated with virus infection. NtEPHX is responsive to TMV infection, but there is no evidence that NtEPHX is involved in the resistance of TMV infection [40]. Our result showed that NbLeuC, not NbEPHX3, was indispensable for RSV infection. The silence of NbLeuC might interfere with the biosynthesis of L-leucine, and then interfere with the protein biosynthesis of RSV. In contrast to NbLeuC and NbEPHX3, NbDCP1 might be ubiquitinated and degraded during RSV infection. As a component of the decapping complex, DCP1 participated in the degradation of mRNAs [41]. It was reported that the silencing of NbDCP1 promoted the infection of turnip mosaic virus in *N. benthamiana* [42]. In this study, the silence of NbDCP1 only promoted RSV symptom at the late stage, not in the early stage. This reminds us that NbDCP1 might have multiple functions during the whole process of RSV infection, such as the cap acquisition during RSV transcription at the early stage [43]. In the late stage of infection, it may participate in host defense against RSV over-accumulation.

Interestingly, all of seven proteins that coded by RSV were identified to possess one or several Kub sites. To the best of our knowledge, none of these Kub sites has been identified previously. Immunoprecipitation assays showed that ubiquitination modified bands were found for NS2, NS3 and CP proteins. Combined the molecular weight of these bands with the number of ubiquitination sites identified on these three proteins, we speculate that the ubiquitination on these three proteins might not be polyubiquitination. Only one Kub site was identified in RdRp. The DUB activity of the OTU domain might have played an important role in the DUB of RdRp [31]. Future work will investigate the functional mode of ubiquitination modification in RSV-encoded proteins.

## 5. Conclusions

Ubiquitination is widely involved in the physiological processes of RSV infection in *N. benthamiana*. *N. benthamiana* regulates the ubiquitination of host proteins in response to the infection of RSV, on the other hand, *N. benthamiana* directly mediates the ubiquitination of viral proteins. The results of our study improve our understanding of the roles of ubiquitination in RSV infection and are helpful for on understanding of RSV pathogenesis.

## Figures and Tables

**Figure 1 viruses-14-02349-f001:**
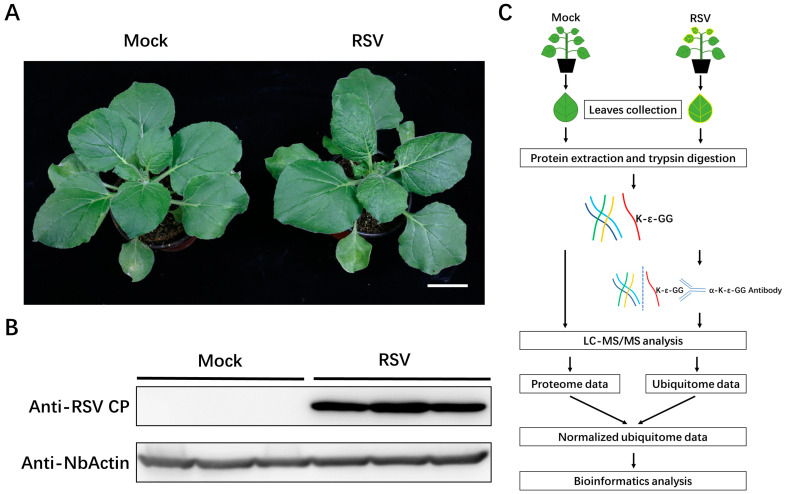
Virus infection and experimental design. (**A**) *N. benthamiana* inoculated with RSV infected rice sap or with healthy rice sap (mock) at 10 dpi. Bar, 3 cm. (**B**) Western blot showing the effective infection of RSV. NbActin was used as a loading control. (**C**) A workflow diagram for the identification and relative quantification of Kub sites in RSV infected or mock *N. benthamiana*.

**Figure 2 viruses-14-02349-f002:**
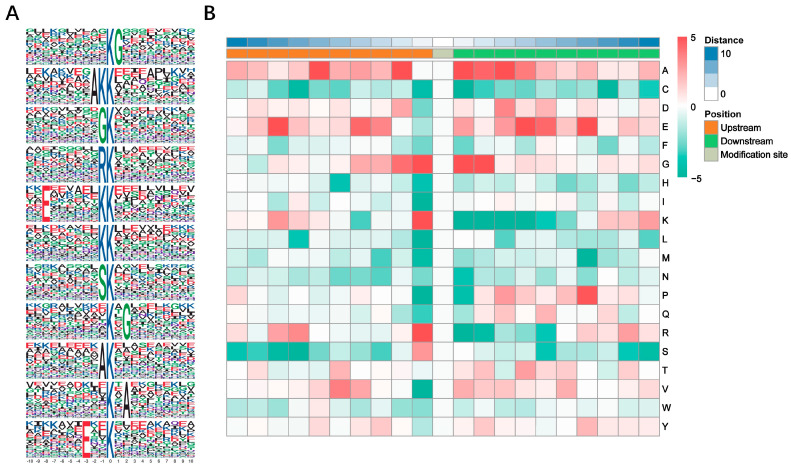
Sequence analysis of the ubiquitinated proteins. (**A**) Enriched ubiquitination motif logos generated by Motif-x. Letters with higher size represent higher frequency of amino acid residues. The central K refers to the ubiquitinated lysine. (**B**) Heat map of the amino acids up- and down-stream of central Kub sites.

**Figure 3 viruses-14-02349-f003:**
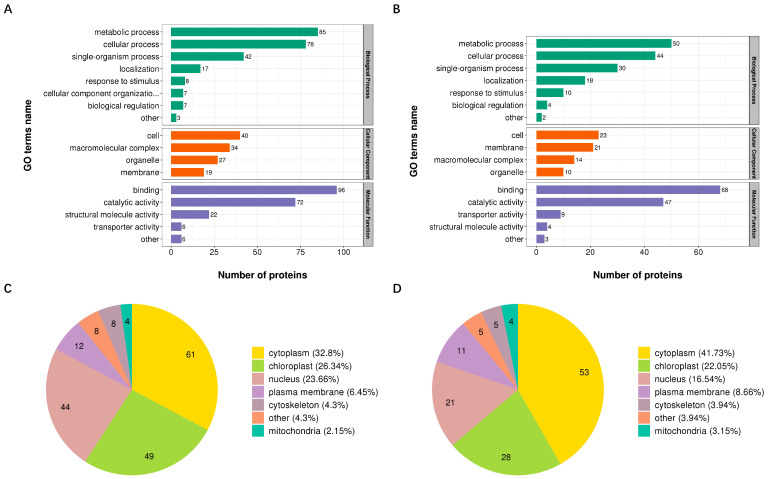
Classification of the identified proteins with DUSs. (**A**,**B**) GO annotation of proteins with up-regulated Kub sites (**A**) and down-regulated Kub sites (**B**) in “biological process category”, “cellular component category”, and “molecular function category”. (**C**,**D**) Subcellular localization of proteins with up-regulated Kub sites (**C**) and down-regulated Kub sites (**D**). Numbers in the sector diagrams represent the number of proteins that possess corresponding subcellular localization.

**Figure 4 viruses-14-02349-f004:**
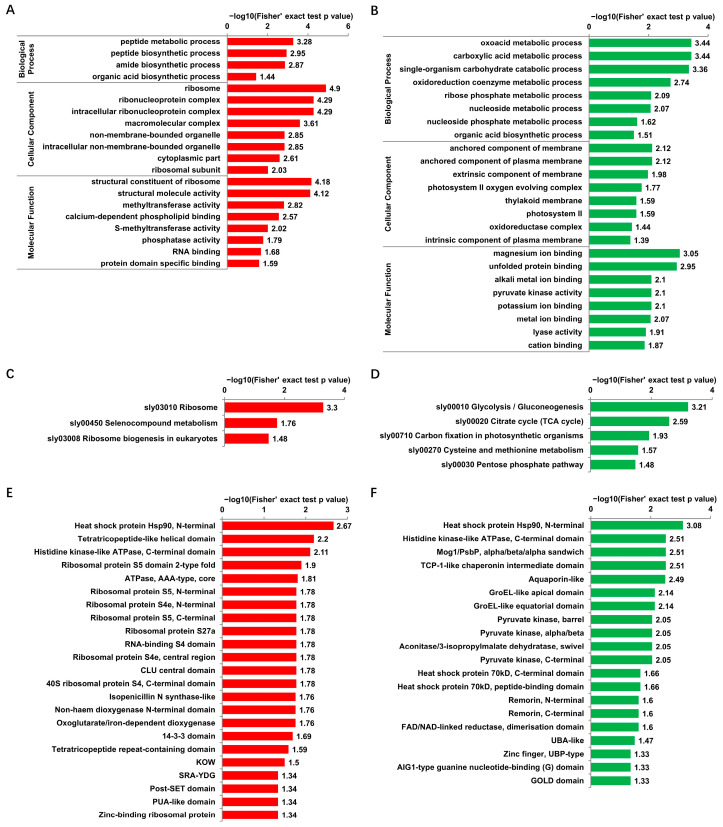
Functional enrichment analysis of identified proteins with DUSs. (**A**,**B**) GO enrichment analysis of proteins with up-regulated Kub sites (**A**) and down-regulated Kub sites (**B**) in “biological process category”, “cellular component category”, and “molecular function category”. (**C**,**D**) KEGG enrichment analysis of proteins with up-regulated Kub sites (**C**) and down-regulated Kub sites (**D**). (**E**,**F**) Protein domain enrichment analysis of proteins with up-regulated Kub sites (**E**) and down-regulated Kub sites (**F**).

**Figure 5 viruses-14-02349-f005:**
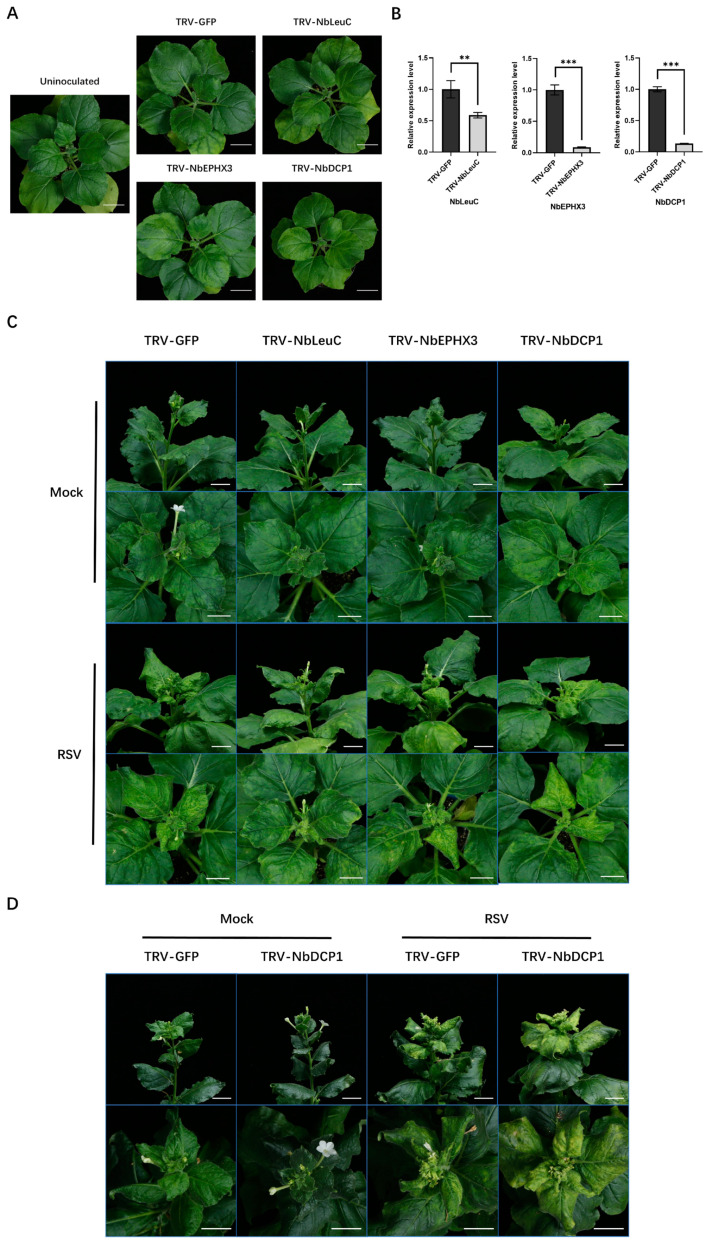
Functions analysis of proteins with DUSs during RSV infection. (**A**) The growth state of healthy *N. benthamiana* and *N. benthamiana* infected with TRV-GFP, TRV-NbLeuC, TRV-NbEPHX3, TRV-NbDCP1 at 10 dpi. Bars, 3 cm. (**B**) RT-qPCR analysis of the silencing efficiency of NbLeuC, NbEPHX3 and NbDCP1 in TRV-NbLeuC, TRV-NbEPHX3 and TRV-NbDCP1 pre-inoculated *N. benthamiana*. The total RNA of the leaf samples from TRV-GFP, TRV-NbLeuC, TRV-NbEPHX3 and TRV-NbDCP1 pre-inoculated *N. benthamiana* plants were extracted for RT-qPCR analyses at 10 dpi. NbActin served as an internal reference in relative quantification. The values represent the means of the expression levels ± SD relative to the TRV-GFP control plants (n = 3 biological replicates). Data were analyzed by Student’s *t* test, and asterisks denote significant differences between TRV-GFP and TRV-NbLeuC, TRV-NbEPHX3 or TRV-NbDCP1 pre-inoculated plants (two-tailed, ** *p* < 0.01, *** *p* < 0.001). (**C**) The viral symptom comparison between *N. benthamiana* plants pre-inoculated with TRV-GFP (control) and TRV-NbLeuC, TRV-NbEPHX3 or TRV-NbDCP1 at 15 dpi. Bars, 2 cm. (**D**) The viral symptom comparison between *N. benthamiana* plants pre-inoculated with TRV-GFP (control) and TRV-NbDCP1 at 30 dpi. Bars, 2 cm.

**Figure 6 viruses-14-02349-f006:**
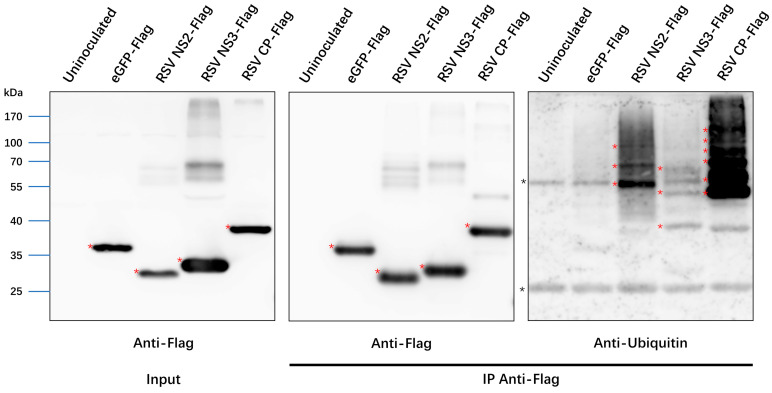
Immunoprecipitation of ubiquitinated RSV proteins in *N. benthamiana*. eGFP-Flag, RSV-NS2-Flag, RSV-NS3-Flag and RSV-CP-Flag were transiently expressed in *N. benthamiana* leaves, respectively. Anti-Flag magnetic beads were used to affinity enrich proteins with Flag tag. Antibody against Flag tag was used to analyze samples before (Input) and after (IP) immunoprecipitation. The bands with red asterisks on the left two images represent the corresponding protein bands. Antibodies against ubiquitin were used to analyze samples after IP. The red asterisks on the right image represent ubiquitinated RSV proteins. Black asterisks referred to the heavy and light chains of beads. The molecular weight was indicated on the left.

**Table 1 viruses-14-02349-t001:** Overview of proteins and Kub sites identified and quantified in proteome and ubiquitome data.

		Identified	Quantified
Proteome	Proteins	5519	3549
Ubiquitome	Sites	2530	1345
	Proteins	1469	862

**Table 2 viruses-14-02349-t002:** Overview of proteins and Kub sites regulated during RSV infection in proteome and normalized ubiquitome data.

	Regulated Type	Fold Change > 1.5
Proteome RSV/Mock	Up-regulated	323 proteins
	Down-regulated	276 proteins
Normalized ubiquitome RSV/Mock	Up-regulated	244 sites
	Down-regulated	155 sites

**Table 3 viruses-14-02349-t003:** Detailed information of chosen proteins with DUSs.

Protein	Position	Ubiquitome RSV/Mock Ratio	Ubiquitome RSV/Mock *p* Value	Proteome RSV/Mock Ratio	Proteome RSV/Mock *p* Value	Normalized Ubiquitome RSV/Mock Ratio	Normalized Ubiquitome RSV/Mock *p* Value
LeuC	K701	0.096	1.82 × 10^−5^	1.51	5.89 × 10^−5^	0.086	6.09 × 10^−6^
EPHX3	K222	0.637	1.08 × 10^−1^	3.288	7.81 × 10^−3^	0.129	7.03 × 10^−4^
DCP1	K138	1.998	1.41 × 10^−4^	0.512	1.25 × 10^−2^	3.896	6.3 × 10^−5^

**Table 4 viruses-14-02349-t004:** Overview of the RSV proteins identified in the proteome data and the identified Kub sites in RSV proteins.

RSV Proteins	Sequence Coverage	Identified Kub Sites
RdRp	2.5%	1270
NS2	67.8%	16, 18, 87
NSvc2	24.7%	42, 243, 300, 351, 353, 575
NS3	89.1%	87, 112, 127, 138, 157, 177, 184, 193
CP	78.3%	5, 21, 88, 90, 214, 231, 246, 256, 289, 292, 300
SP	52.8%	51, 52, 136, 144, 147, 158, 161, 162, 177
MP	37.8%	13, 25, 33, 39, 243, 252, 279

## Data Availability

The mass spectrometry proteome data and ubiquitome data have been deposited to the ProteomeXchange Consortium via the PRIDE partner repository with the dataset identifier PXD037417 and PXD037442.

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
