# Peer review of "Comprehensive Analysis of Ubiquitome Changes in Nicotiana benthamiana after Rice Stripe Virus Infection"

_viruses, 2022, doi:10.3390/v14112349_

Round 1

Reviewer 1 Report

In the present manuscript Liu et al systematically profiled the global changes of the ubiquitination in plants during RSV infection by combining the ubiquitomics and proteomics approach. These data improve our understanding of the roles of ubiquitination in RSV infection and sheds light on understanding of RSV pathogenesis. This work will have broad interest and provide significant progress in the study of plant-virus interaction. This is an interesting, thorough, and well-written paper. There are some minor errors should be corrected before acceptance for publication.

1, In Figure 5A, the most left graph is not “wild type”, since all these tobacco were wild type, it should be changed to “uninoculated” or “untouched”. Similarly, in Figure 6, “Nb WT” should be changed to “uninoculated”.

2. Please enlarge Fig. 4, it is too small to see the words in the graph.

3. Can the authors describe how many peptides from the viral proteins were identified? Or Can the authors show the coverage of the peptides on the viral genome?

Author Response

1, In Figure 5A, the most left graph is not “wild type”, since all these tobacco were wild type, it should be changed to “uninoculated” or “untouched”. Similarly, in Figure 6, “Nb WT” should be changed to “uninoculated”.

Our response: Revised accordingly.

2. Please enlarge Fig. 4, it is too small to see the words in the graph.

Our response: We put all the words in bold to make it easier to be read.

3. Can the authors describe how many peptides from the viral proteins were identified? Or Can the authors show the coverage of the peptides on the viral genome?

Our response: Web thank the reviewer for the valuable suggestion. We described the coverage of the peptides on the viral genome that identified in the proteome data in the results and Table 4

Reviewer 2 Report

The manuscript entitled: ’Comprehensive analysis of ubiquitome changes in Nicotiana benthamiana after rice stripe virus infection’ raise a very interesting topic which need to be elucidated, however, the way the manuscript was prepared is deficient. The work has no aim and no hypothesis was set. The Material and Methods section doesn’t provide sufficient detail to allow others to replicate the results. In the Discussion section the obtained results from bioinformatic analyses weren’t discussed and confronted with the available literature.

Introduction section:

No aim and hypothesis of the work was mentioned here, as  well as, the main conclusions weren’t highlighted.

Materials and methods section:

First, there is too many paragraphs. Please combine some of them. For example, 2.3. paragraph ‘Isolation of samples’, in which a word ‘isolation’ is incorrect and should be changed to ‘harvest’ or ‘collection’, can be added to 2.1. paragraph “Plant materials…’. Paragraphs 2.9., 2.10., and 2.11. can be combined into one bioinformatic analysis-describing paragraph.

The paragraphs aren’t arranged in ‘chronological order’ or the provided data is not full. For example, in ‘Plasmid construction’ paragraph:

83-84: How total RNA was obtained?

84-85: What DNA fragments were cloned? What were the conditions of the PCR? What primers were used and based on what sequences these were designed?

There is no information concerning the time points for collecting material for the analysis such as Western blot or RT-qPCR. In the plant material paragraph, it should be specified at what time point the material was collected for specific analysis.

There is almost any software information that was used in the mentioned bioinformatic analyses.

Moreover, there is too much ‘…as described previously’, which makes this section incomplete and chaotic. Please briefly describe the cited methods.

Discussion section:

As no aim and hypothesis were made, no reference to this hypothesis in the discussion was taken. Authors discuss only the results concerning the three chosen proteins: NbLeuC, NbEPHX3, and NbDCP1. Authors didn’t write about the results of the GO enrichment analysis or the identified pathways that are related to the up- or down-regulated DUFs.

Conclusions:

Authors write only about the work that was done. No results were mentioned. These are not conclusions.

Authors contributions:

The contribution of YW and JW wasn’t specified.

Data availability:

According to the Instructions for Authors: ‘All generated mass spectrometry raw data must be deposited in the appropriate public database such as ProteomeXchange, PRIDE or jPOST.’ Please provide accession number for all proteomic data.

Line by line comments:

30: RSV is not a negative virus but negative-sense and ambisense single-stranded RNA virus. Please change.

32-33: What are other ways of RSV transmission?

47: ‘to mediates’ should be ‘to mediate’

52: put abbreviation for Argonaute1

59-63: This information is not suitable for the Introduction section.

64-66: The development of mass spectrometry is associated with proteomic but not with transcriptomics. Please rephrase

87: ‘in planta’ should be written in italics

88, 92: restriction enzymes names should be written in italics

89-93: Based on what sequences the primers were designed? It should be added to the Table S1.

90: Please, explain the abbreviation

97, 170: leaves cannot be isolated in this meaning, they can be ‘collected’ or ‘harvested’. RNA or protein can be isolated from leaves.

211-219: Authors wrote about proteins classification into three GO categories but didn’t describe the results of this analysis.

254-256: What is the justification for choosing those three proteins for VIGS?

272-275: What is the explanation for RbcL accumulation analysis?

294-300: Please explain in the text more precisely what is shown on the Figure 6, especially on the panel with ANTI-Ubiquitin. Why so many bands are seen in the each lane with virus proteins? Please add a marker. Also, in the ANTI-Ubiquitin panel a positive band is shown, while authors claim that ubiquitin could not be precipitated with eGFP-Flag. How the reader is supposed to know which the positive bands are and which don’t?

314: ‘in vitro’ should be written in italics

323, 325: ‘creatures’ is an inappropriate word, change the word

482: this is not reference

Figures:

There is an inconsistency with the figure labelling, in the figures there are capital letters (e.g., 1A), while in the figure legends – small letters (e.g., 1(a)).

Figures 3 and 4 are hard to read and must be highly enhanced to be seen properly.

Figure 6: Marker is lacking and in the panel with ANTI-Ubiquitin it is not known what all these bands stands for. Which bands are artifacts and which are the ubiquinated proteins?  

Figure S1 B and C: positive bands for ANTI-RSV CP are shown in the controls in the western blotting – author must provide a figure with negative results in the control.

This work has multiple English language mistakes, including wrong word usage in the context of their meaning, which wasn’t mentioned above, hence the manuscript must go through English language and style editing.

Author Response

Reviewer 2

The manuscript entitled: ’Comprehensive analysis of ubiquitome changes in Nicotiana benthamiana after rice stripe virus infection’ raise a very interesting topic which need to be elucidated, however, the way the manuscript was prepared is deficient. The work has no aim and no hypothesis was set. The Material and Methods section doesn’t provide sufficient detail to allow others to replicate the results. In the Discussion section the obtained results from bioinformatic analyses weren’t discussed and confronted with the available literature.

Introduction section:

No aim and hypothesis of the work was mentioned here, as  well as, the main conclusions weren’t highlighted.

Our response: Revised accordingly. 

Materials and methods section:

First, there is too many paragraphs. Please combine some of them. For example, 2.3. paragraph ‘Isolation of samples’, in which a word ‘isolation’ is incorrect and should be changed to ‘harvest’ or ‘collection’, can be added to 2.1. paragraph “Plant materials…’. Paragraphs 2.9., 2.10., and 2.11. can be combined into one bioinformatic analysis-describing paragraph.

Our response: Revised accordingly.

The paragraphs aren’t arranged in ‘chronological order’ or the provided data is not full. For example, in ‘Plasmid construction’ paragraph:

83-84: How total RNA was obtained?

Our response: We put the information on RNA isolation as suggested.

84-85: What DNA fragments were cloned? What were the conditions of the PCR? What primers were used and based on what sequences these were designed?

Our response: Revised accordingly. The names of DNA sequences and primers used are provided as suggested.

There is no information concerning the time points for collecting material for the analysis such as Western blot or RT-qPCR. In the plant material paragraph, it should be specified at what time point the material was collected for specific analysis.

Our response: Revised accordingly.

There is almost any software information that was used in the mentioned bioinformatic analyses.

Moreover, there is too much ‘…as described previously’, which makes this section incomplete and chaotic. Please briefly describe the cited methods.

Our response: Revised accordingly. 

Discussion section:

As no aim and hypothesis were made, no reference to this hypothesis in the discussion was taken. Authors discuss only the results concerning the three chosen proteins: NbLeuC, NbEPHX3, and NbDCP1. Authors didn’t write about the results of the GO enrichment analysis or the identified pathways that are related to the up- or down-regulated DUFs.

Our response: We describe the reason why select these three genes. The results of the GO enrichment analysis or the identified pathways that are related to the up- or down-regulated DUFs were discussed.

Conclusions:

Authors write only about the work that was done. No results were mentioned. These are not conclusions.

 Our response: Revised accordingly.

Authors contributions:

The contribution of YW and JW wasn’t specified.

 Our response: Revised accordingly.

Data availability:

According to the Instructions for Authors: ‘All generated mass spectrometry raw data must be deposited in the appropriate public database such as ProteomeXchange, PRIDE or jPOST.’ Please provide accession number for all proteomic data.

Our response: We are uploading the data, and will put the information in proof stage.

Line by line comments:

30: RSV is not a negative virus but negative-sense and ambisense single-stranded RNA virus. Please change.

Our response: Revised accordingly.

32-33: What are other ways of RSV transmission?

Our response: Revised accordingly.

47: ‘to mediates’ should be ‘to mediate’

Our response: Revised accordingly.

52: put abbreviation for Argonaute1

Our response: Revised accordingly.

59-63: This information is not suitable for the Introduction section.

Our response: These sentences were deleted accordingly.

64-66: The development of mass spectrometry is associated with proteomic but not with transcriptomics. Please rephrase

Our response: Revised accordingly.

87: ‘in planta’ should be written in italics

Our response: Revised accordingly.

88, 92: restriction enzymes names should be written in italics

Our response: Revised accordingly.

89-93: Based on what sequences the primers were designed? It should be added to the Table S1.

Our response: Revised accordingly.

90: Please, explain the abbreviation

Our response: Revised accordingly.

97, 170: leaves cannot be isolated in this meaning, they can be ‘collected’ or ‘harvested’. RNA or protein can be isolated from leaves.

Our response: Revised accordingly.

211-219: Authors wrote about proteins classification into three GO categories but didn’t describe the results of this analysis.

Our response: Revised accordingly.

254-256: What is the justification for choosing those three proteins for VIGS?

Our response: Revised accordingly.

272-275: What is the explanation for RbcL accumulation analysis?

Our response: Revised accordingly.

294-300: Please explain in the text more precisely what is shown on the Figure 6, especially on the panel with ANTI-Ubiquitin. Why so many bands are seen in the each lane with virus proteins? Please add a marker. Also, in the ANTI-Ubiquitin panel a positive band is shown, while authors claim that ubiquitin could not be precipitated with eGFP-Flag. How the reader is supposed to know which the positive bands are and which don’t?

Our response: Revised accordingly.

314: ‘in vitro’ should be written in italics

Our response: Revised accordingly.

323, 325: ‘creatures’ is an inappropriate word, change the word

Our response: Revised accordingly.

482: this is not reference

Our response: We deleted this reference.

Figures:

There is an inconsistency with the figure labelling, in the figures there are capital letters (e.g., 1A), while in the figure legends – small letters (e.g., 1(a)).

Our response: Revised accordingly.

Figures 3 and 4 are hard to read and must be highly enhanced to be seen properly.

Our response: Revised accordingly. We renewed the Fig. 3 and put all the words in the Fig. 4 in bold to make it easier to be read.

Figure 6: Marker is lacking and in the panel with ANTI-Ubiquitin it is not known what all these bands stands for. Which bands are artifacts and which are the ubiquinated proteins?  

Our response: Revised accordingly.

Figure S1 B and C: positive bands for ANTI-RSV CP are shown in the controls in the western blotting – author must provide a figure with negative results in the control.

Our response: Revised accordingly.

This work has multiple English language mistakes, including wrong word usage in the context of their meaning, which wasn’t mentioned above, hence the manuscript must go through English language and style editing.

Our response: We carefully checked the manuscript man times to reduce the mistakes.

Round 2

Reviewer 2 Report

The Authors answered to the comments properly , however, there are some issues that still must be improved.

As omics analyses, including ubiquitomics, provide a large number of data, a table presenting the fold changes for all proteins with DUSs after normalization and the corresponding protein level should be supplied as Supplementary Material. The table should contain the protein accession number, protein name, ubiquitination positions, fold change, and P value. The only information concerning N. benthamiana proteins with DUSs during RSV infection was provided only for 3 proteins, while 399 DUSs in 313 proteins were identified.

Additional, line by line comments:

Table S1: Please provide the accession numbers for the tested genes (those with coding sequences).

77: ‘…and various cellular components/pathways after RSV infection.'  It should be during RSV infection

118: Explain RbcL abbreviation.

131: A sentence should not start with a number.

149: Please explain the abbreviation for NSI.

150: Please provide a full producers name.

156: Please provide N. benthamiana database version.

156-157: Please provide the source of RSV sequences. Are these sequences deposited in GenBank? Provide the accession numbers.

163: Please provide Perseus software version.

174: Stick to the names spelling: Gene Ontology (GO), Kyoto Encyclopedia of Genes and Genomes (KEGG)

265-268: It should be, ‘In the biological process category,…’, ‘In the cellular component category,…’, ‘In the molecular function category,…’. The same should be applied to the Figure 3 and Figure 4 legends.

339: RbcL abbreviation was introduced before.

369-373: This is not a properly written sentence, rephrase.

403: '...biological materials suffering a series of treatments...' - rephrase this statement

443-448: These sentences are not written grammatically, rephrase. The bands were found for…

Author Response

As omics analyses, including ubiquitomics, provide a large number of data, a table presenting the fold changes for all proteins with DUSs after normalization and the corresponding protein level should be supplied as Supplementary Material. The table should contain the protein accession number, protein name, ubiquitination positions, fold change, and P value. The only information concerning N. benthamiana proteins with DUSs during RSV infection was provided only for 3 proteins, while 399 DUSs in 313 proteins were identified.

Our responses: As suggested, a Table S2 is included (The original Table S2 becomes Table S3)

 Additional, line by line comments:

 Table S1: Please provide the accession numbers for the tested genes (those with coding sequences).

Our responses: Revised accordingly.

77: ‘…and various cellular components/pathways after RSV infection.'  It should be during RSV infection

118: Explain RbcL abbreviation.

131: A sentence should not start with a number.

 149: Please explain the abbreviation for NSI.

150: Please provide a full producers name.

Our responses: The above parts were revised accordingly.

156: Please provide N. benthamiana database version.

Our responses: Database version is included.

156-157: Please provide the source of RSV sequences. Are these sequences deposited in GenBank? Provide the accession numbers.

Our responses: The accession numbers are provided.

163: Please provide Perseus software version.

 Our responses: Revised accordingly.

174: Stick to the names spelling: Gene Ontology (GO), Kyoto Encyclopedia of Genes and Genomes (KEGG)

Our responses: Revised accordingly.

265-268: It should be, ‘In the biological process category,…’, ‘In the cellular component category,…’, ‘In the molecular function category,…’. The same should be applied to the Figure 3 and Figure 4 legends.

Our responses: Revised accordingly.

339: RbcL abbreviation was introduced before.

 Our responses: Revised accordingly.

369-373: This is not a properly written sentence, rephrase.

403: '...biological materials suffering a series of treatments...' - rephrase this statement

443-448: These sentences are not written grammatically, rephrase. The bands were found for…

Our responses: The above parts were rephrased.
